# Folliculin promotes substrate-selective mTORC1 activity by activating RagC to recruit TFE3

Kristina Li[1,2☉], Shogo Wada[1☉], Bridget S. Gosis[1], Chelsea Thorsheim[1], Paige Loose[1], Zolt Arany[1] *

1 Cardiovascular Institute, Perelman School of Medicine, University of Pennsylvania, Pennsylvania, United States of America, 2 Department of Bioengineering, University of Pennsylvania, Pennsylvania, United States of America

☉ These authors contributed equally to this work.
* zarany@pennmedicine.upenn.edu

## Abstract

Mechanistic target of rapamycin complex I (mTORC1) is central to cellular metabolic regulation. mTORC1 phosphorylates a myriad of substrates, but how different substrate specificity is conferred on mTORC1 by different conditions remains poorly defined. Here, we show how loss of the mTORC1 regulator folliculin (FLCN) renders mTORC1 specifically incompetent to phosphorylate TFE3, a master regulator of lysosome biogenesis, without affecting phosphorylation of other canonical mTORC1 substrates, such as S6 kinase. FLCN is a GTPase-activating protein (GAP) for RagC, a component of the mTORC1 amino acid (AA) sensing pathway, and we show that active RagC is necessary and sufficient to recruit TFE3 onto the lysosomal surface, allowing subsequent phosphorylation of TFE3 by mTORC1. Active mutants of RagC, but not of RagA, rescue both phosphorylation and lysosomal recruitment of TFE3 in the absence of FLCN. These data thus advance the paradigm that mTORC1 substrate specificity is in part conferred by direct recruitment of substrates to the subcellular compartments where mTORC1 resides and identify potential targets for specific modulation of specific branches of the mTOR pathway.

## Introduction

The ability of a cell to sense and respond to the intracellular and extracellular environment is vital for it to maintain metabolic homeostasis. Doing so is also fundamentally necessary for the cell to align its metabolic programming to ongoing cellular physiological needs. A major component of sensory integration occurs at the mechanistic target of rapamycin complex I (mTORC1) kinase complex [1–5]. This multisubunit complex integrates numerous inputs, including signals from growth factors, ambient levels of various amino acids (AAs), the cellular energy state, and hypoxia and DNA damage. In turn, it regulates multiple metabolic programs, for example, promoting anabolic processes such as lipid and protein synthesis, while inhibiting catabolic processes such as autophagy and lysosome biogenesis [1–5].

**Data Availability Statement:** All relevant data are within the paper and its Supporting Information files.

**Funding:** SW was supported by a postdoctoral fellowship from the American Diabetes

Association, BG was supported by the National Institutes of Health (NIH) (F30) and the Blavatnik Family Foundation, and ZA was supported by the NIH (R01 DK107667). The funders had no role in study design, data collection and analysis, decision to publish, or preparation of the manuscript.

**Competing interests:** The authors have declared that no competing interests exist.

**Abbreviations:** AA, amino acid; BHD, Birt–Hogg–Dubé; dFBS, dialyzed FBS; DMEM, Dulbecco's Modified Eagle Medium; FLCN, folliculin; GAP, GTPase-activating protein; gRNA, guide RNA; LOH, loss of heterozygosity; mTORC1, mechanistic target of rapamycin complex I; RCC, renal cell carcinoma; WT, wild type.

The mTORC1 complex, nucleated around the adaptor protein Raptor, is recruited to the lysosome membrane upon AA sufficiency and then activated by Rheb in response to growth factors, achieved by relieving the repression of Rheb by the TSC complex [1–5]. AA sensing by mTORC1 is complex, including sensing of leucine by Sestrin and sensing of arginine by SLC38A9. In response to these integrated inputs, mTOR phosphorylates a myriad of targets, including p70S6K and 4EBP1 to promote protein translation and ribosome biogenesis, ULK1 to suppress autophagy, Lipin1 to promote lipid synthesis, and the TFE3/B transcription factors to suppress lysosome biogenesis [1–5]. The mTORC1 pathway is thus often depicted as monolithic, acting as a single on/off switch that senses dozens of upstream informational inputs and integrates them into the single response of phosphorylating its multiple targets [1–5]. However, such a monochromatic model of central control of cellular homeostasis is highly unlikely to be accurate.

We have recently identified a substrate-specific branch of mTORC1 signaling, providing the first example of specific regulation of different branches of mTORC1 signaling [6,7], subsequently also reported by the Zoncu and Ballabio groups [8,9]. In this pathway, the protein folliculin (FLCN) regulates mTORC1-mediated phosphorylation of only TFE3/B, while not affecting phosphorylation of other canonical substrates such as S6K and 4EBP1. Thus, deletion of FLCN completely abrogates phosphorylation of TFE3, releasing it from 14-3-3 binding and cytoplasmic sequestration and allowing its nuclear translocation to drive genes of lysosome and mitochondria biogenesis. In contrast, deletion of FLCN does not disable phosphorylation of canonical substrates like S6K and 4EBP1 [6,7]. Understanding how, mechanistically, FLCN confers this substrate specificity onto the mTORC1 complex is thus of significant interest.

FLCN is a GTPase-activating protein (GAP) and thus stimulator of the small G-proteins RagC and D, which are active in their GDP-bound state [10]. RagC and D heterodimerize with RagA or B to incorporate into the mTORC1 complex and positively regulate mTORC1 activity. Structures elucidated by cryoEM reveal FLCN to bind directly to RagC/D [8,11], confirming earlier coprecipitation studies [12], and prior work has indicated that RagC binds to TFE3 [13]. We thus hypothesized here that the mechanism by which FLCN modulates only the TFE3/B arm of mTORC1 signaling is by activating RagC to recruit TFE3 to the mTORC1 complex, i.e., achieving substrate specificity via specific recruitment of substrate to the complex. While the work that we report here was being finalized, the Ballabio group reported overlapping findings with TFEB [9].

## Results

### TFE3 phosphorylation is responsive to AAs, via the GATOR complex

To begin to investigate the specific regulation of TFE3 phosphorylation, we tested the impact on TFE3 phosphorylation by known upstream regulators of canonical mTORC1 activity: growth factors and AAs. C2C12 cells were grown in complete media containing growth factor-rich 10% fetal bovine serum (FBS). The cells were then changed for 60 minutes into either complete media, media lacking AAs but containing dialyzed FBS (dFBS), media with AAs but no FBS, or media with neither. As seen in the "NTC" columns of Fig 1A, phosphorylation of TFE3 at S320, the mTORC1-targeted site, detected with a phospho-specific antibody, was seen in full media and media lacking serum, but not in media lacking AAs (quantification in S1 Fig). Consistent with its dephosphorylation, TFE3 translocated to the nucleus in the absence of AAs (Fig 1B, "NTC"). Thus, TFE3 phosphorylation depends more on AA sensing than on growth factor sensing.

Canonical mTORC1 signaling senses growth factor signals via the inactivation of the repressive TSC complex and senses the presence of leucine via inactivation of the repressive

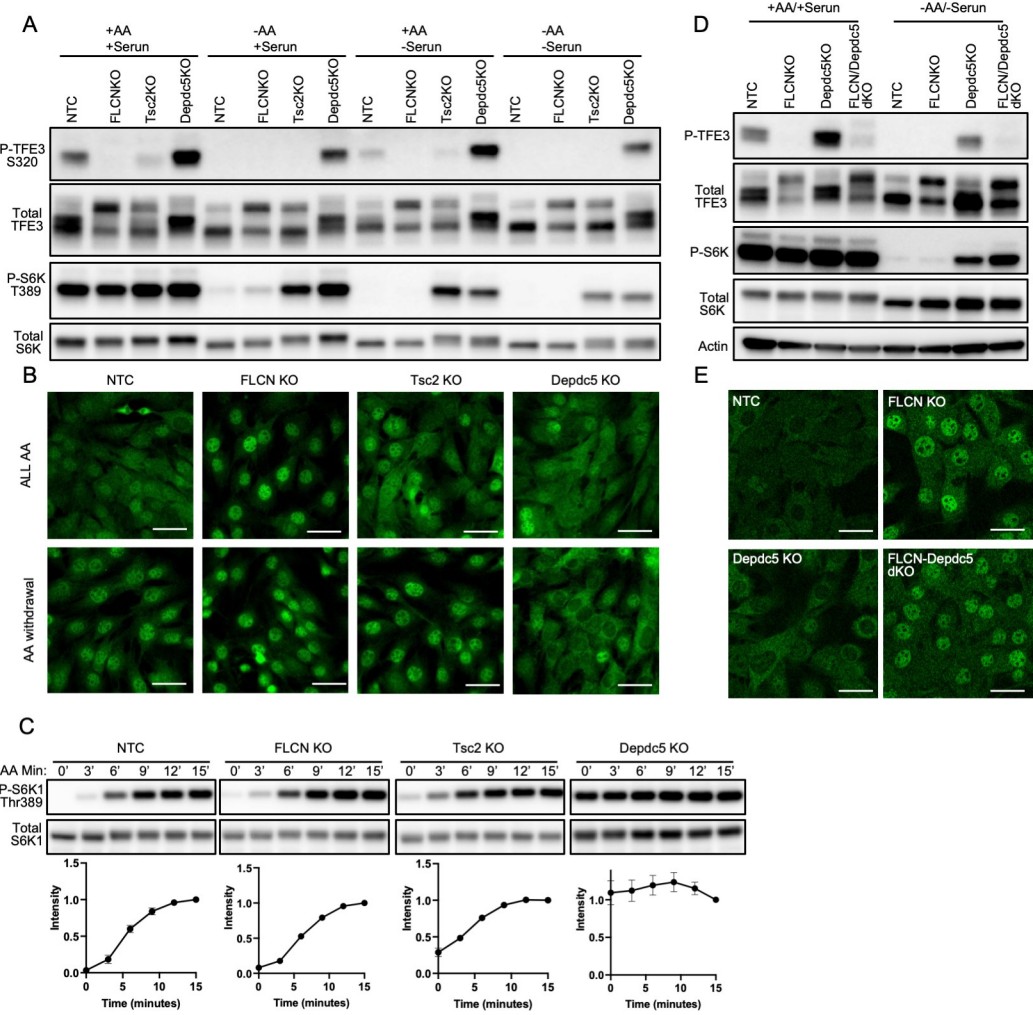

**Fig 1. TFE3 phosphorylation is responsive to AAs, via the GATOR complex. (A, B)** Control C2C12 cells (NTC) or cells lacking Flcn, Tsc2, or Depdc5, as indicated, were switched from complete medium to media lacking serum and/or AAs, as indicated, for 60 minutes, followed by immunoblotting for TFE3, phospho-TFE3, S6K, and phospho-S6K (A) or immunohistochemistry for subcellular localization of TFE3 (B). **(C)** The same cells as in A, after 60 minutes in medium lacking AAs, were returned to complete media for the indicated time points and immunoblotted for S6K, and phospho-S6K. Values were normalized to the 15-minute time point of each line. **(D, E)** C2C12 cells lacking Flcn, Depdc5, or both, as indicated, were evaluated as in A and B. The data underlying all the graphs shown in the figure is included in S1 Data. AA, amino acid; FLCN, folliculin; KO, knockout.

GATOR1 complex [1]. Thus, CRISPR/Cas-9–mediated deletion of either *Tsc2* (an obligatory component of TSC) or of *Depdc5* (an obligatory component of GATOR1) led to constitutive phosphorylation of the canonical target S6K, even in the absence of AAs or serum (Fig 1A, "Tsc2KO" and "Depdc5KO"). In contrast, only deletion of *Depdc5* led to constitutive phosphorylation of TFE3, while deletion of *Tsc2* did not. These data lead to 2 conclusions: First, growth factor signaling via TSC inhibition cannot promote phosphorylation of TFE3, thus separating canonical and noncanonical signals. Second, phosphorylation of TFE3 in response to AAs is mediated in large part via DEPDC5.

We have shown previously that FLCN regulates the phosphorylation of TFE3 [6,7], as first described by the Linehan group [14]. Consistent with this, under all conditions tested, cells lacking *Flcn* also lacked any detection of TFE3 phosphorylation (Fig 1A, "FlcnKO"). In

contrast, deletion of *Flcn* had no impact on S6K phosphorylation under any of the conditions tested. To investigate if FLCN may have a specific role in the kinetics of S6K phosphorylation in response to AAs, we also performed a time course after AA replenishment (Fig 1C). At no time point, however, was S6K phosphorylation altered in the cells lacking Flcn (Fig 1C). We conclude that AA sensing is intact in cells lacking *Flcn* and that FLCN is entirely dispensable for canonical AA signaling to S6K, again separating canonical and noncanonical signals. Of note, while concordant with our prior observations in other cell types [6,7,9], these findings differ from those of Tsun and colleagues [10], perhaps reflecting our use of complete knockout via CRISPR, in contrast to the siRNA approach taken by Tsun and colleagues.

Finally, to test if AA sensing via GATOR promotes TFE3 phosphorylation via FLCN, we generated cells lacking both *Depdc5* and *Flcn* (dKO cells). As seen in Fig 1D, *Flcn* was epistatic to *Depdc5*, i.e., loss of *Depdc5* failed to activate phosphorylation of TFE3 in the absence of FLCN, whether in the presence or absence of AAs and serum. Together, these data demonstrate that the presence of AAs is necessary and sufficient to promote phosphorylation of TFE3 by mTORC1, and does so via GATOR1 and FLCN, while growth factor signaling via TSC inhibition does not promote phosphorylation of TFE3, in sharp contrast to canonical phosphorylation of S6K.

## RagC, but not RagA, promotes TFE3 phosphorylation in response to AAs

FLCN is a GAP for the highly similar Rags C and D, either of which heterodimerizes with either RagA or B to activate mTORC1 in response to AAs. RagC and D are active in the GDP-bound form, while RagA and B are active in the GTP-bound form. To test which Rag type primarily drives TFE3 phosphorylation, we expressed in C2C12 or 293T cells HA-tagged wild type (WT) or constitutively active RagA (66L mutant, mimicking GTP-bound state) or RagC (75L, mimicking GDP-bound state) [15]. In the absence of AAs, neither WT construct was able to rescue phosphorylation of either S6K (canonical signal) or TFE3 (noncanonical) (Fig 2A, S2A Fig). Constitutively active RagA (66L) efficiently reactivated phosphorylation of S6K in C2C12 cells (but not in 293T cells, reflecting cell-specific effects), while having little impact on TFE3 phosphorylation (Fig 2A). In sharp contrast, constitutively active RagC (75L) reactivated TFE3 phosphorylation in both cell types, while having no impact on S6K, despite the relatively lower expression of RagC 75L protein (Fig 2A, S2A Fig). The latter in part reflected higher protein instability of RagC 75L protein, as revealed by treating cells with the proteasome inhibitor MG132 (S2B Fig). Coexpression of RagC 75L and RagA 66L had no impact on TFE3 phosphorylation beyond that conferred by RagC 75L alone (S2A Fig). Consistent with these findings, only RagC 75L promoted cytoplasmic sequestration of TFE3 in the absence of AA (Fig 2B).

To evaluate the kinetics of this process, we treated cells with complete media, switched the cells to media lacking AAs and then evaluated dephosphorylation of TFE3, S6K, and 4EBP serially over 120 minutes (Fig 2C, S2C Fig). Removal of AAs led to 90% dephosphorylation of TFE3 by 30 minutes, and 80% dephosphorylation of S6K and 4EBP by 45 minutes. Expression of constitutively active RagC 75L delayed dephosphorylation on TFE3, while having little impact on S6K and 4EBP. Conversely, expression of constitutively active RagA 66L largely maintained phosphorylation on S6K and 4EBP, while having little impact on TFE3. The latter is consistent with the observation in Fig 1D that activation of RagA, conferred by deletion of its inhibitor DEPDC5, is not sufficient to rescue TFE3 phosphorylation in the absence of FLCN. These data thus demonstrate clearly separable Rag-mediated pathways, whereby RagA is sufficient to promote canonical signaling to S6K and 4EBP in response to AA stimulation, while RagC is sufficient to promote signaling to TFE3 without simultaneous additional activation of RagA beyond its preexisting baseline activity.

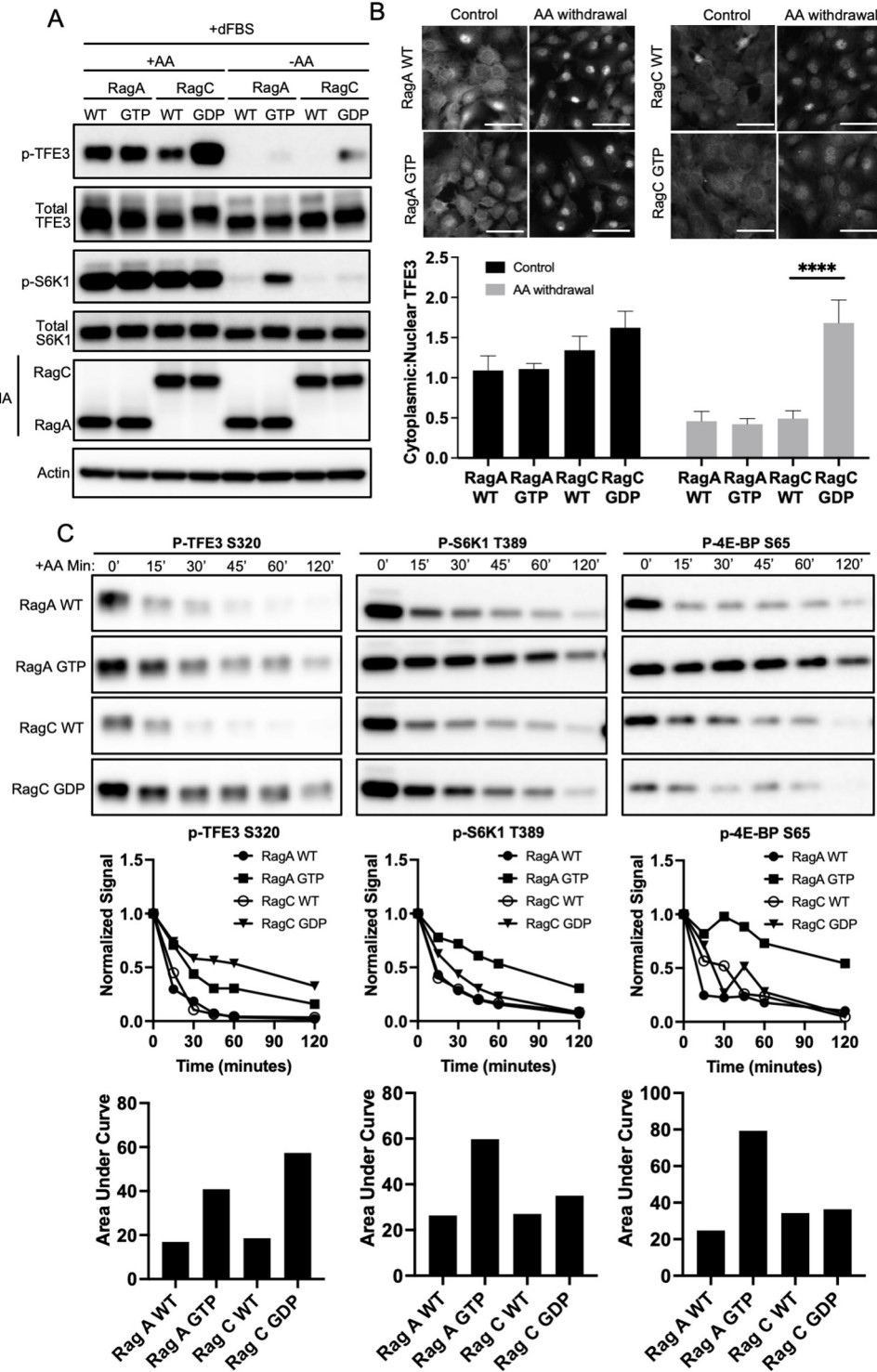

**Fig 2. RagC, but not RagA, promotes TFE3 phosphorylation in response to AAs. (A, B)** C2C12 cells expressing HA-tagged WT or constitutive active RagA (GTP) or RagC (GDP) were switched from complete medium to media lacking AAs, as indicated, followed by immunoblotting for TFE3, phospho-TFE3, S6K, phospho-S6K, and HA (A) or immunohistochemistry for subcellular localization of TFE3 (B). Quantification of cytoplasmic-to-nuclear ratio of TFE3 is shown below the images. Scale bar: 20 μm, ****$p < 0.0001$ by Student $t$ test. **(C)** The same cells as in A, subjected to a time course after withdrawal of AAs, followed by immunoblotting for phospho-TFE3, phospho-4EBP, and phospho-S6K. Densitometric quantification is shown below. The data underlying all the graphs shown in the figure is included in S1 Data. AA, amino acid; dFBS, dialyzed FBS; WT, wild type.

## Active RagC rescues TFE3 phosphorylation in the absence of FLCN, while active RagA does not

To test if RagC confers specificity on noncanonical signaling to TFE3, WT and constitutively active RagA and C were expressed in C2C12 cells lacking *Flcn* and grown in complete media (Fig 3A). Despite the presence of AAs, TFE3 remained unphosphorylated, reflecting the absence of *Flcn*. Strikingly, only expression of RagC 75L could rescue phosphorylation of TFE3 in these cells (Fig 3A). Consistent with this, only RagC 75L could promote cytoplasmic sequestration of TFE3 in FLCN knockout cells (Fig 3B). When activated and nuclear, TFE3 is known to drive a broad genetic program, including upregulating expression of lysosome proteins

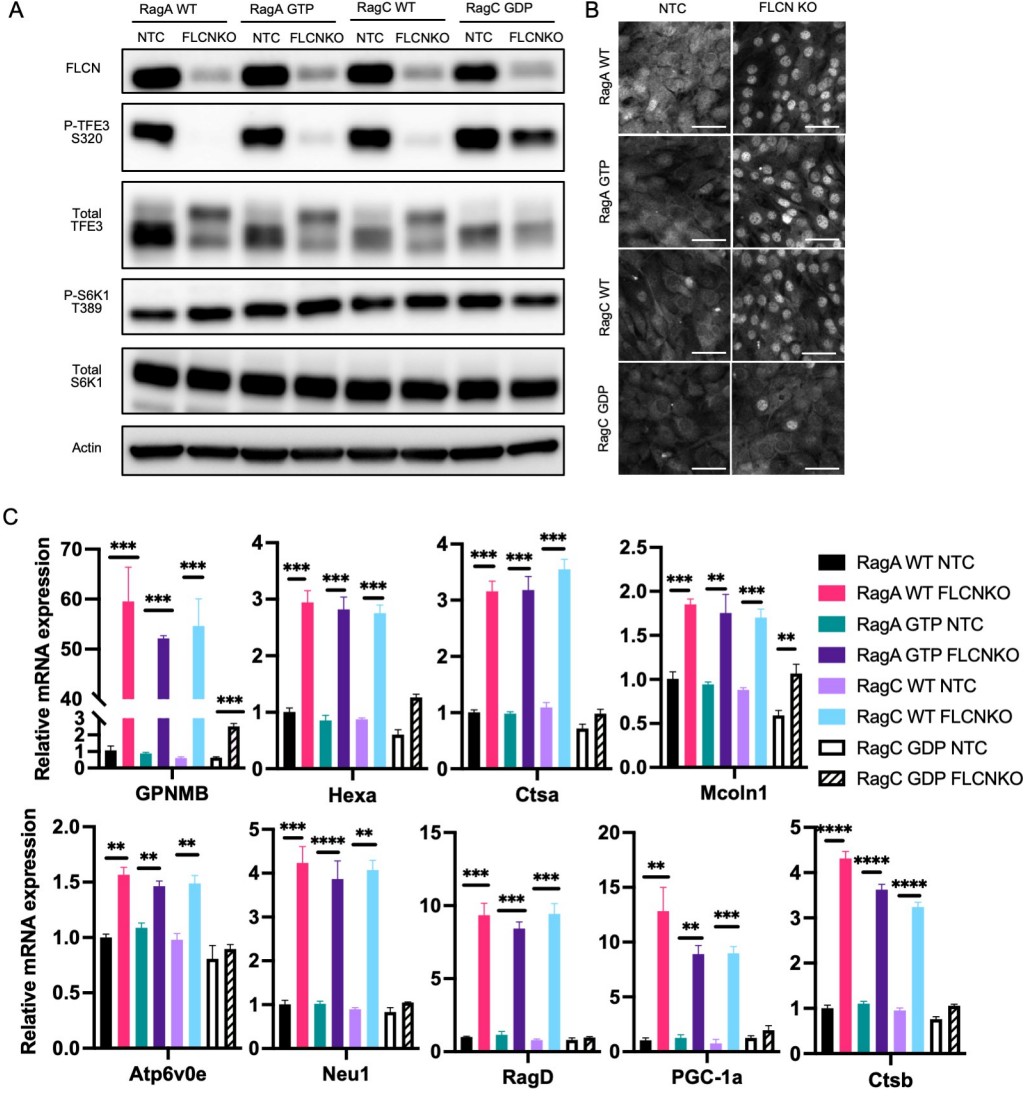

**Fig 3. Constitutively active RagC, but not RagA, rescues TFE3 phosphorylation in the absence of FLCN. (A, B)** Control C2C12 cells and cells lacking *Flcn* were transduced with HA-tagged WT or constitutive active RagA (66L) or RagC (75L), followed by immunoblotting for FLCN, TFE3, phospho-TFE3, S6K, and phospho-S6K (A), immunohistochemistry for subcellular localization of TFE3 (B), or quantitative PCR evaluation of expression of the indicated genes (normalized to the average expression of HPRT, TBP, and 36B4 as controls) **(C)**. Scale bar: 10 μm, $^{**}p < 0.01$, $^{***}p < 0.001$, $^{****}p < 0.0001$ by Student *t* test. The data underlying all the graphs shown in the figure is included in S1 Data. FLCN, folliculin; KO, knockout; WT, wild type.

(*Mcoln1*, *Neu1*, *Hexa*, *Atp6v0e*, *Ctsa*, *Ctsb*, and *Gpnmb*) [13], regulators of mitochondrial bio-genesis (*Ppargc1a*) [16], and a positive feedback loop to mTORC1 via *Ragd* [7,17]. All of these genes were dramatically induced in the absence of FLCN (Fig 3C). Furthermore, only the expression of RagC 75L prevented their induction, consistent with the phosphorylation and cytoplasmic sequestration of TFE3 (Fig 3C). Thus, we conclude that RagC is epistatic to FLCN, i.e., that FLCN promotes TFE3 phosphorylation via RagC and that RagC confers substrate specificity to the mTORC1 complex.

## AA stimulation drives transient localization of TFE3 to lysosome via FLCN and RagC

To begin to evaluate how RagC confers substrate specificity to the mTORC1 complex, we evaluated TFE3 subcellular localization in response to AA stimulation. Puertollano's group reported that, in nutrient-replete cells, TFE3 transiently translocates to the lysosome, where it is phosphorylated by mTORC1, followed by binding to 14-3-3 and cytoplasmic sequestration [13]. The transient translocation to the lysosome can be captured by inhibiting mTORC1 activity with Torin1 (Fig 4A, top panel, costaining TFE3 with LAMP2, a lysosome marker). Strikingly, in cells lacking *Flcn*, TFE3 entirely fails to translocate to the lysosome (Fig 4A, bottom panel). Thus, FLCN serves the critical function of recruiting substrate (TFE3) to mTORC1. Equally strikingly RagC 75L entirely rescued the translocation of TFE3 to the lysosome in cells lacking *Flcn* (Fig 3B). These data demonstrate that activation of RagC, occurring physiologically via FLCN GAP activity, recruits TFE3 to the lysosome, leading to its phosphorylation and cytoplasmic retention.

## RagC is necessary and sufficient for AA-stimulated TFE3 localization to lysosome and subsequent phosphorylation

The data above demonstrated the sufficiency of activated RagC to drive TFE3 lysosome localization and phosphorylation. To test if RagC is required for this process, we generated by CRISPR/Cas-9 C2C12 cells lacking RagC (S3A Fig). These cells revealed a near complete block of TFE3 phosphorylation in response to AA stimulus (Fig 5A). In clear contrast, canonical

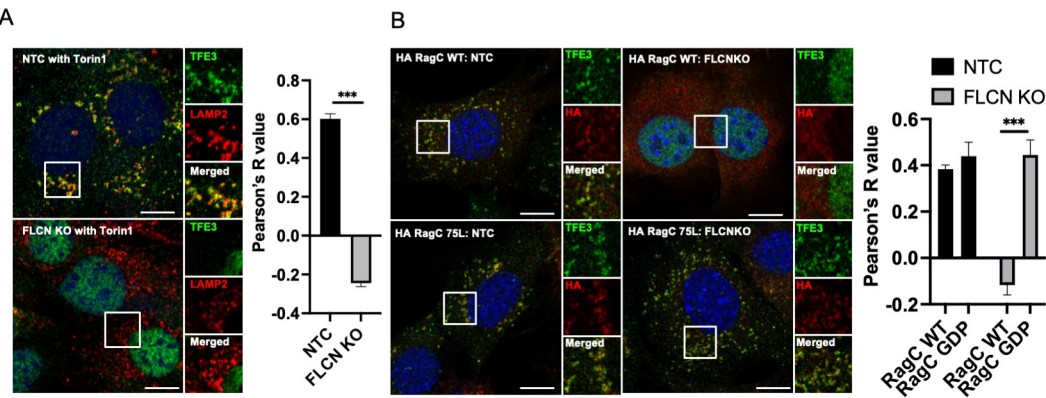

**Fig 4. AA stimulation drives transient localization of TFE3 to lysosome via FLCN and RagC. (A)** Control C2C12 cells and cells lacking *Flcn* were maintained for 60 minutes in medium lacking AAs and then returned to complete media in the presence of Torin1 for 15 minutes, followed by immunohistochemistry for subcellular localization of TFE3 and LAMP2, a marker of the lysosome. Right: correlation by Pearson's R of LAMP2 and TFE3 staining. **(B)** Cells lacking *Flcn* were transduced with HA-tagged WT or constitutive active RagA (66L) or RagC (75L), followed by immunohistochemistry as in A. ∗∗∗$p < 0.001$ by Student $t$ test ($n = 3$). Scale bar: 10 μm. The data underlying all the graphs shown in the figure is included in S1 Data. AA, amino acid; FLCN, folliculin; KO, knockout; WT, wild type.

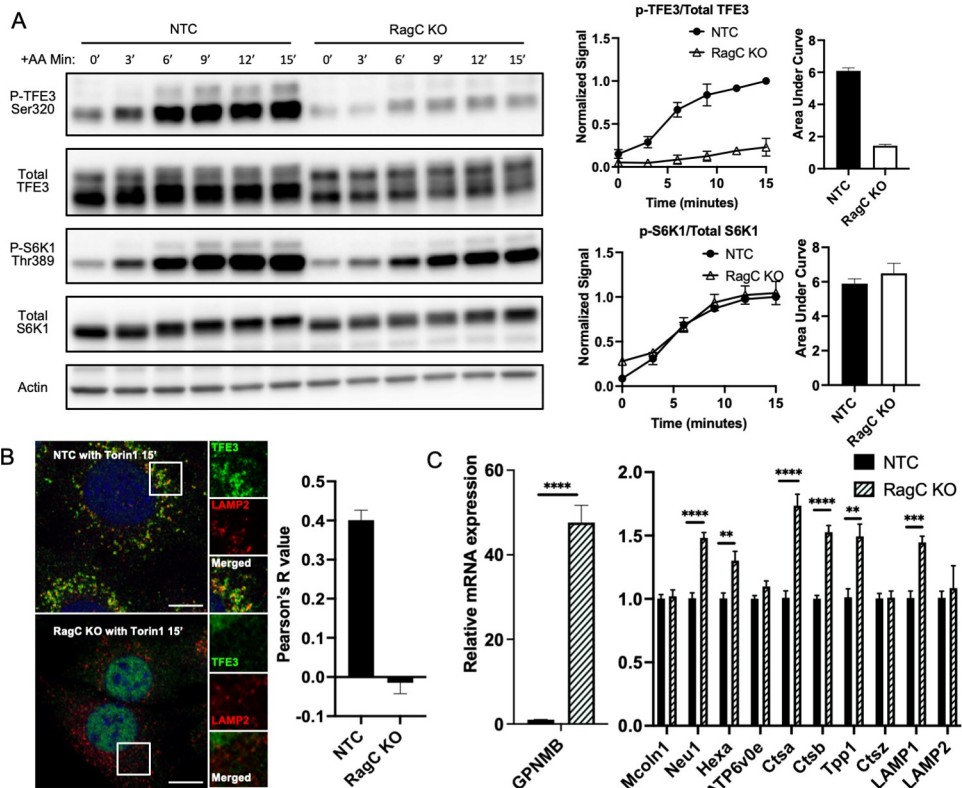

**Fig 5. RagC is necessary for AA-stimulated TFE3 phosphorylation and localization to lysosome. (A)** Control C2C12 cells and cells lacking *RagC* were maintained for 60 minutes in medium lacking AAs and then returned to complete media for the indicated times, followed by immunoblotting as indicated. Phospho-TFE3/totalTFE3 and phospho-S6K/totalS6K were quantified and normalized to the 15-minute time point of NTC (control). **(B)** Control C2C12 cells and cells lacking *RagC* were cultured with 250 nM Torin1 for 15 minutes, followed by immunohistochemistry for subcellular localization of TFE3 and LAMP2, a marker of the lysosome. (Scale bar: 10 μm). On the right: correlation by Pearson's R of LAMP2 and TFE3 staining. **(C)** Quantitative PCR evaluation of expression of the indicated genes (normalized to the average expression of HPRT, TBP, and 36B4 as controls). $^{**}p < 0.01$, $^{***}p < 0.001$, $^{****}p < 0.0001$ by Student $t$ test. The data underlying all the graphs shown in the figure is included in S1 Data. AA, amino acid; KO, knockout.

phosphorylation of S6K in response to AA was entirely unaffected in these RagC knockout cells (Fig 5A). No compensatory induction of RagD was appreciated in the absence of RagC (S3B Fig). Thus RagC is required for AA signaling to TFE3, but dispensable for AA signaling to S6K, clearly separating the 2 arms of mTORC1 signaling. Note that despite being dispensable, there is evidence that overexpression of RagC mutants can suppress S6K phosphorylation, likely working in a dominant-negative fashion [18]. Evaluation of TFE3 subcellular localization in response to AA stimulus revealed that RagC was equally required for the recruitment of TFE3 to the lysosome (Fig 5B). Thus, we find that activation of RagC by FLCN is both necessary and sufficient to recruit TFE3 to the lysosome and to promote its phosphorylation, without simultaneous additional activation of RagA.

## Discussion

The mechanisms by which mTORC1 integrates upstream signals and transmits them downstream has been extensively and elegantly characterized [1]. However, how such a complex integrator of multiple inputs achieves specificity in its outputs has received little attention. We

first demonstrated clearly that one branch of mTORC1 output could be independently regulated from another, i.e., we showed that loss of FLCN, a RagC/D GAP, abrogated mTORC1-mediated phosphorylation of TFE3 while having no impact on canonical phosphorylation of S6K and 4EBP [6,7,19]. The impact in vivo of such selective regulation in different cell types included beiging of adipocytes and chronic activation of monocytes. Lacking from these studies, however, was a clear mechanistic understanding of how FLCN confers substrate specificity on mTORC1. We elucidate here this mechanism of substrate specificity, whereby FLCN activates RagC to its GDP-bound form via its GAP activity; activated RagC then physically recruits TFE3 to lysosome surface, thereby promoting its phosphorylation by mTORC1; phosphorylated TFE3 is then bound to 14-3-3 and sequestered in the cytoplasm, thus suppressing TFE3 target pathway activation in the nucleus. Similar findings were recently reported for the regulation of TFEB [9]. Importantly, these events occur independently of Rheb and RagA-mediated regulation of canonical phosphorylation of S6K and 4EBP. Disruption of GATOR1, a GAP of RagA/B, renders mTORC1 insensitive to AA withdrawal, maintaining TFE3 as well as canonical substrates phosphorylated even in the absence of AAs. Concomitant loss of FLCN selectively blunted TFE3 phosphorylation while phosphorylation status of canonical substrates remains insensitive to AA withdrawal, further supporting the separable branches of mTORC1.

Heterozygous loss-of-function germline mutations in FLCN cause Birt–Hogg–Dubé (BHD) syndrome, which is marked by chronic development of lung cysts, abundant benign dermal hamartoma-like tumors, and a high incidence of renal cell carcinoma (RCC) [20]. Both the dermal tumors and RCC are characterized by high canonical mTORC1 activity and yet occur in the context of loss of heterozygosity (LOH), i.e., loss of FLCN-mediated activation of mTORC1. The existence of the substrate-specific mechanism described here helps to explain this seeming paradox: loss of FLCN unleashes TFE3 to the nucleus, but has no direct impact on canonical mTORC1 signaling. Moreover, as we have shown before [7], an indirect positive feedback loop explains how in some cell types canonical mTORC1 activity in fact increases in the absence of FLCN: Nuclear TFE3 strongly induces gene expression of RagD [17], which can drive canonical mTORC1 phosphorylation of S6K even in the absence of FLCN [7]. The mechanistic separation of mTORC1 signaling into FLCN-independent (canonical) and FLCN-dependent (noncanonical) arms thus explains the apparent paradoxical development of tumors with high mTORC1 activity in BHD patients.

We note evidence of 2 reciprocal feedback loops between these 2 arms of mTORC1 signaling. On the one hand, inactivation of FLCN can lead to RagD-mediated activation of canonical S6K phosphorylation, as described above. Conversely, we also note that constitutive activation of canonical signaling, achieved via deletion of *Tsc2*, leads to reciprocal partial suppression of TFE3 phosphorylation (Fig 1). This observation is consistent with a previous study, in which unbiased genetic screens revealed TSC to act upstream of FLCN and TFE3 in the regulation of exit from pluripotency in embryonic stem cells [21]. The mechanism for this second feedback loop remains unclear.

TFE3 is member of a small family of bHLH-ZIP-type transcription factor that includes TFEB, TFEC, and MITF [22]. Interestingly, TFE3 translocations and gene duplications (i.e., gain-of-function variants) are a relatively common cause of kidney cancer, associated with high mTORC1 activity, thus mimicking the effects of FLCN deletion in BHD syndrome [23]. TFEB and MITF mutations have also been noted in kidney cancers, albeit more rarely. Genetic deletion of *Flcn* in the kidney in mice yields severe polycystic disease, but not frank cancer, indicating that additional genetic hits are likely required to develop cancer. Ballabio's group recently showed that codeletion of *Tfeb* rescues the polycystic phenotype of kidney-specific *Flcn* deletion [9]. In the same study, the authors show similar effects of RagC on TFEB as we show here on TFE3. There is thus likely a fair amount of similarity between TFE3 and TFEB

pathways. The fact that deletion of either *Tfeb* or *Tfe3* abrogates the effect of *Flcn* deletion suggests that TFE3 and TFEB may heterodimerize, although such interaction has not been reported to date. Alternatively, TFE3 and TFEB perform different functions in different tissues, as suggested by, for example, the lethality of whole-body deletion of *Tfeb*, while *Tfe3* knockout mice are viable, with little baseline phenotype [24,25].

In summary, we elucidate here the mechanistic basis by which FLCN confers substrate specificity upon the mTORC1 complex: FLCN activates RagC to physically recruit TFE3 to the mTORC1 complex, promoting TFE3 phosphorylation while having little impact on canonical substrates such as S6K. Our work, combined with similar work with TFEB [9], mechanistically exposes the first clear example of parsing of mTORC1 signaling.

## Materials and methods

### Cell culture

Mouse C2C12 myoblasts were cultured in Gibco Dulbecco's Modified Eagle Medium (DMEM) with high glucose and GlutaMAX (Invitrogen 10569010, MA, USA) supplemented with 10% FBS and 1% penicillin streptomycin (Invitrogen 15140122). Cells were incubated in 37°C and 5% CO2. DMEM media was changed every 2 days and split with trypsin (Invitrogen 25200056) when cells reached 95% confluence.

### Nutrient withdrawal and restimulation experiment

For AA withdrawal experiments, cells were washed with sterile PBS and AA-free DMEM with 10% dFBS was placed on cells for specified times. For restimulation experiments, AA free media was replaced with complete media (DMEM with 10% FBS).

### Antibodies

Phospho-TFE3 (Ser320) antibody was a gift from Dr. Rosa Puertollano and previously described [13]. Other antibodies used are as follows: total TFE3 (Cell Signaling Technology, 14779, MA, USA), phospho-p70S6K (Thr389) (Cell Signaling Technology, 9234), HA tag (Cell Signaling Technology, 2367), LAMP2 (Abcam, ab13524, MA, USA), FLCN (Abcam, ab124885), total p70S6K (Cell Signaling Technology, 2708), beta-actin (Cell Signaling Technology, 4970), and 14-3-3 (Cell Signaling Technology, 8312)

### Gene deletion by CRISPR/Cas-9 system

lentiCRISPR version 2 was a gift from Feng Zhang (Department of Biological Engineering, Massachusetts Institute of Technology, Cambridge, Massachusetts, USA) (Addgene, plasmid 52961). The guide RNAs (gRNAs) were designed using the Optimized CRISPR Design website (http://crispr.mit.edu) from Zhang Lab (Cambridge, MA, USA). Mouse C2C12 myoblast cells were infected with lentivirus encoding for the Cas-9/gRNA, selected using puromycin, and validated using western blot before being used as populations. The gRNA sequence was as follows (the pam sequence is excluded): mouse nontarget control (5′-ATTGTTCGACCGTC TACGG G-3′), mouse flcn (5′-TCCGTGCAGAAGAGCGTGCG-3′), mouse tsc2 (5′-TTGATGCAATG TATTCGTCA-3′), mouse depdc5 (5′-GACAAGTTTGTAGACCTTTG-3′), and mouse RagC (5′-GGACTTCGGCTACGGCGTGG-3′).

### Lenti and retro virus production

Lenti transfer plasmid, psPAX2 (Addgene, plasmid 12260), and pMD2.G (Addgene, plasmid 12259) (both gifts from Didier Trono, School of Life Sciences, EcolePolytechnique Federale de

Lausanne, Lausanne, Switzerland) were cotransfected onto HEK293T cells using Lipofectamine 3000 (Thermo Fisher Scientific, MA, USA). Plasmids were removed after 18 hours on HEK293T cells, and media was replenished. After 48 hours, conditioned media was collected and passed through a low protein binding 0.45-μm syringe filter to remove cell debris. Mouse C2C12 myoblasts were plated on 6-well multiwell plates and infected via spinfection method. Polybrene was added to the virus containing media (final concentration of 8 μg/mL) and added on top of the C2C12 cells. Each 6-well plate was centrifuged for 90 minutes at 1,000 g (spinfection), and media was replaced. Stable cells were selected 24 hours postspinfection with the appropriate antibiotic.

## Western blot

Cell culture samples were lysed with RIPA buffer with a proteinase inhibitor (Complete mini-proteinase inhibitor cocktail, Roche, BS, CH) and a phosphatase inhibitor (PhosSTOP, Roche). Samples were sonicated and spun down to remove lipid and insoluble debris. A BCA protein assay kit (Thermo Fisher Scientific) was used to quantify and normalize protein concentrations. The same amount of protein (10 to 20 μg) was loaded on to a 4% to 20% gradient Tris-glycine polyacrylamide gel (Bio-Rad, CA, USA) and electrophoresed (SDS-PAGE). Samples were transferred to PVDF membrane (MilliporeSigma, MA, USA) and blocked with 5% milk for 1 hour and incubated with primary antibody overnight. The following day, membranes were washed with TBS-T and incubated in appropriate HRP-conjugated secondary antibody for 60 minutes. Images were taken using the ImageQuant LAS 4000 (GE Healthcare Life Sciences, NJ, USA).

## Immunoprecipitation

Cells were washed with cold PBS and lysed with IP Lysis Buffer (Thermo Fisher Scientific) with proteinase inhibitor (Complete miniproteinase inhibitor cocktail, Roche) and a phosphatase inhibitor (PhosSTOP, Roche). Samples were sonicated and centrifuged. Supernatant was added to Pierce Anti-HA Magnetic Beads (Thermo Fisher Scientific) and incubated at room temperature for 30 minutes with end-over-end mixing. Samples were washed 3 times with TBS-T, reconstituted in 1xSDS RIPA, and boiled at 95°C for 5 minutes. HA beads were removed.

## Immunocytochemistry

Cells were grown on glass coverslips precoated with collagen type I. The following day, cells were fixed with 4% paraformaldehyde (Thermo Fisher Scientific) for 15 minutes. Cells were washed with PBS and incubated in blocking buffer (1XPBS/5% Normal Goat Serum/0.3% Triton X-100) for 60 minutes at room temperature. Primary antibodies were diluted as indicated on respective datasheets in blocking buffer and incubated overnight on cells at 4°C. Cells were washed with PBS and incubated in appropriate Alexa Flour (Thermo Fisher Scientific) secondary antibodies diluted in blocking buffer for 1 hour at room temperature in the dark. After washing with PBS, coverslips were mounted with Prolong Gold Antifade Reagent with DAPI (Cell Signaling Technology). Images were captured using the Zeiss LSM 710 (BW, DE) confocal microscope, and image analysis was done using ImageJ.

## Supporting information

**S1 Fig. TFE3 phosphorylation is responsive to AAs, via the GATOR complex. (A)** Control C2C12 cells (NTC) or cells lacking Flcn, Tsc2, or Depdc5 were switched from complete medium to media lacking serum and/or AAs for 60 minutes followed by immunoblotting.

Images were uploaded into ImageJ, and signal intensity was quantified. Graphed above is the ratio of p-TFE3 to total TFE3 signal. **(B)** Quantification and graph of ratio of pS6K to total S6K signal. The data underlying all the graphs shown in the figure is included in S1 Data. AA, amino acid; FLCN, folliculin.
(PDF)

**S2 Fig. RagC, but not RagA, promotes TFE3 phosphorylation in response to AAs. (A)** Coexpression of active RagA and RagC in 293T cells does not confer further phosphorylation of TFE3 compared to active RagC alone. **(B)** RagC 75L protein is partially stabilized by inhibition of the proteasome with MG132. **(C)** To accompany main Fig 2C, immunoblotting for total levels of TFE3, S6K1, and 4E-BP in C2C12 cells expressing HA-tagged WT, or constitutive active RagA (GTP) or RagC (GDP), demonstrate equivalent expression of these proteins at all time points after switching from complete medium to media lacking AAs. AA, amino acid; WT, wild type.
(PDF)

**S3 Fig. RagD expression is not increased as compensation for RagC CRISPR KO or siRNA knockdown. (A)** C2C12 cells with RagC CRISPR KO were used for Fig 5A. **(B)** RagC CRISPR KO (in C2C12s) and RagC siRNA knockdown (in HEK 293Ts) showed no significant compensation of RagD expression. The data underlying all the graphs shown in the figure is included in S1 Data. KO, knockout.
(PDF)

**S1 Data. Data underlying graphs in main and Supporting information figures.**
(XLSX)

**S1 Raw Images. Raw images.**
(PDF)

## Author Contributions

**Conceptualization:** Kristina Li, Shogo Wada, Bridget S. Gosis, Zolt Arany.

**Data curation:** Kristina Li, Shogo Wada, Bridget S. Gosis, Chelsea Thorsheim, Paige Loose, Zolt Arany.

**Formal analysis:** Kristina Li, Shogo Wada, Bridget S. Gosis, Chelsea Thorsheim, Zolt Arany.

**Funding acquisition:** Zolt Arany.

**Investigation:** Kristina Li, Shogo Wada, Bridget S. Gosis, Chelsea Thorsheim, Paige Loose, Zolt Arany.

**Methodology:** Kristina Li, Shogo Wada, Bridget S. Gosis, Zolt Arany.

**Project administration:** Shogo Wada, Zolt Arany.

**Resources:** Zolt Arany.

**Supervision:** Zolt Arany.

**Writing – original draft:** Kristina Li, Shogo Wada, Zolt Arany.

**Writing – review & editing:** Kristina Li, Shogo Wada, Bridget S. Gosis, Zolt Arany.

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
