## [Editor Report · Decision Letter 0]

4 Mar 2021

Dear Zolt, 

Thank you for submitting your manuscript entitled "FLCN promotes substrate-selective mTORC1 activity by activating RagC to recruit TFE3 to the lysosome" for consideration as a Research Article by PLOS Biology. Thank you also for your patience as we completed our editorial process, and apologies for the delay.

Your manuscript has now been evaluated by the PLOS Biology editorial staff as well as by an academic editor with relevant expertise and I am writing to let you know that we would like to send your submission out for external peer review.

Please re-submit your manuscript within two working days, i.e. by Mar 08 2021 11:59PM.

Kind regards,

Ines

--

Ines Alvarez-Garcia, PhD,

Senior Editor

PLOS Biology

---

## [Decision Letter · Decision Letter 1]

13 May 2021

Dear Dr Arany,

Thank you very much for submitting your manuscript "FLCN promotes substrate-selective mTORC1 activity by activating RagC to recruit TFE3 to the lysosome" for consideration as a Research Article at PLOS Biology. Thank you also for your patience as we completed our editorial process, and please accept my apologies for the delay in providing you with our decision. Your manuscript has been evaluated by the PLOS Biology editors, an Academic Editor with relevant expertise, and by two independent reviewers.

As you will see, the reviews think that the conclusions are interesting and potentially significant for the field, but they also raise several concerns that would need to be addressed to confirm the findings. Both reviewers suggest several experiments to strengthen the results – including using a different cell line - mention several relevant references that you should add to give proper credit to related literature and ask you to clarify several points.

In light of the reviews (attached below), we will not be able to accept the current version of the manuscript, but we would welcome re-submission of a much-revised version that takes into account the reviewers' comments. We cannot make any decision about publication until we have seen the revised manuscript and your response to the reviewers' comments. Your revised manuscript is also likely to be sent for further evaluation by the reviewers.

We expect to receive your revised manuscript within 3 months. 

**IMPORTANT - SUBMITTING YOUR REVISION**

3. Resubmission Checklist

a) *Published Peer Review*

b) *PLOS Data Policy*

Please provide the data underlying the following figures, and make sure you mention in the corresponding figure legends WHERE THE DATA CAN BE FOUND. Please also ensure that your Data Statement in the submission system accurately describes where your data can be found.

Sincerely,

Ines

--

Ines Alvarez-Garcia, PhD

Senior Editor

PLOS Biology

Reviewers’ comments

Rev. 1:

This manuscript by Wada et al. describes how FLCN promotes TFE3 recruitment to the lysosome by activating RagC. TFE3 belongs to the MiT-TFE family of transcription factors, which also includes MITF, TFEB, and TFEC. Previous studies have shown that these transcription factors are regulated in a similar manner, through a mechanism that involves mTORC1-mediated control of nucleo-cytoplasmic shuttling. More recent studies by several groups, including the group of authors of the present manuscript, have shown that mTORC1-mediated regulation of TFEB and TFE3 occurs through a "non-canonical" pathway that requires the FLCN-RagC/D axis. In the present manuscript the authors show that an mTORC1 substrate-specific mechanism, which was previously shown for TFEB (PMID: 32612235), also operates for TFE3. This observation is interesting and potentially relevant, considering the evidence that these two transcription factors may have different roles and relevance in both physiological and pathological settings. However, there are several issues, both in the text and figures, that need to be addressed by the authors.

CREDIT TO PREVIOUS STUDIES

The authors tend to ignore or minimize crucial contributions that other groups have previously made to the specific topic of the present manuscript. The following issues must be addressed by acknowledging previous discoveries and properly citing relevant previous papers.

1) A recent manuscript described a substrate-specific mechanism by which mTORC1 differentially phosphorylates its substrates and how this mechanism is relevant for Birt-Hogg-Dube' syndrome (PMID: 32612235). However, the results and conclusions of this previous manuscript have been largely ignored by the authors. Here are some examples: a) In the abstract (line 27) the authors stated "mTORC1 phosphorylates a myriad of substrates, but how different substrate specificity is conferred on mTORC1 by different conditions is largely unknown". This sentence does not take into account the mechanism recently described in ref. PMID: 32612235. b) Line 80. The hypothesis made by the authors at the end of the introduction is strikingly identical to the main conclusions of ref. PMID: 32612235. It looks like the authors ignored the important findings of this study by hypothesizing an identical mechanism… c) Line 291. The authors claim to have elucidated a new mechanism that enables substrate-specific phosphorylation by mTORC1. However, once again, they seem to ignore that this mechanism was previously described for TFEB (PMID: 32612235). d) Lines 306-314. The "BHD mTORC1 hyperactivity paradox" was recently explained in reference PMID: 32612235 . e) Line 344. The sentence "The work mechanistically exposes the first example of parsing of mTORC1 signaling" is incorrect, as it does not account for the mTORC1 substrate-specific mechanism described in ref. PMID: 32612235.

2) Line 66. Although the authors were the first to report the evidence of a "non-canonical" mTORC1 pathway that mediates TFE3 phosphorylation in adipose and myeloid tissue, additional studies reported differential phosphorylation of TFE3 versus canonical mTORC1 substrates in FLCN deficient cells (PMID: 21209915, PMID: 31672913).

3) Line 110. The first evidence that FLCN regulates the phosphorylation of TFE3 was described in ref. PMID: 21209915. This reference should be added.

4) Line 169. It has been well established by multiple studies that FLCN acts upstream of RagC (PMID: 24095279, PMID: 24081491, PMID: 31672913, PMID: 31704029). Such studies should be cited and the sentence "To test if RagC acts upstream or downstream of FLCN..." should be eliminated.

5) Line 177. The first evidence of an mTORC1-TFEB/TFE3-RagC/D feedback loop was reported in reference PMID: 28619945. Reference 7 only refers to the genes that are regulated by TFE3. Thus, reference PMID: 28619945 should be added when referring to the feedback loop (see also discussion Line 309).

TECHNICAL AND CONCEPTUAL ISSUES

6) Based on the data shown in Figure 2, the authors conclude that activation of RagA is required for the phosphorylation of S6K, but dispensable for the phosphorylation of TFE3. However, this conclusion cannot be drawn from the data shown in this manuscript for the following reasons: in Fig 2A the levels of TFE3 and S6K phosphorylation in starved cells expressing either active RagA or C are very weak. To claim that RagA activity is not relevant for TFE3 phosphorylation, the authors should compare the levels of TFE3 phosphorylation in cells expressing active RagC only, with those observed in cells expressing both active RagA and C. In Fig 2B a staining for RagA/C should be used to show which cells express active RagC/A. Furthermore, based on data in Fig 2B the authors claim that active RagC, but not active RagA, promotes cytosolic sequestration of TFE3 in the absence of AA. However, from these panels it appears to me that amino acid withdrawal induces TFE3 nuclear translocation in both active RagA- and active RagC-expressing cells. The authors should also add quantifications for these IF data. In Figure 2C loading controls are missing, making quantifications unreliable. RagA/C immuno-blotting data are also missing. My overall impression from Fig 2C is that both active RagA and active RagC show a very similar effect on TFE3 phosphorylation in starved cells. In addition to the aforementioned technical issues, the statement that activation of RagA is dispensable for TFE3 phosphorylation conflicts with the data in Figure 1A showing constitutive phosphorylation of TFE3 in cells lacking DEPDC5, a specific inhibitor of RagA/B.

7) Line 221. The authors state that "RagC is dispensable for AA signaling to S6K". However, Rag GTPases are known to work as dimers of either RagA or B bound to RagC or D. No published evidence suggests that RagA/B could work as monomers. The authors' conclusion that RagC is dispensable for S6K phosphorylation is based on the data in Fig 5a showing that phosphorylation of S6K is only marginally affected in RagC-KO cells. However, RagC and RagD are well known to be able to compensate for each other. Therefore, in Fig 5A the authors should assess the levels of RagD (as well as those of RagC), to determine whether a possible RagD compensation mechanism exists in RagC-KO cells. In addition, to claim that RagC is dispensable for S6K phosphorylation, the authors should assess the phosphorylation of S6K in a RagC/D-double KO cell line or in RagC-KO cells silenced for RagD.

8) Based on the data of Figure 6 the authors claim that anchoring TFE3 to different cytosolic compartments rescues its phosphorylation in FLCN-deficient cells. The mechanism by which such rescue occurs is unclear. In addition Figure 6 contains several important issues that need to be addressed (see below).Alternatively, the authors may consider eliminating this figure from the paper. Points to be addressed in Figure 6:

Fig6A: the levels of TFE3-Rheb15 in FLCN-KO cells are huge compared to the levels of either WT-TFE3 or TFE3-CAAX observed in the same cells (compare lane 4 with lanes 2 and 6). With such a huge difference in expression levels, it is impossible to come to any conclusion on the phosphorylation status of the different constructs in FLCN-KO cells.

Fig6E: Similarly, the levels of deltaNLS-TFE3 in FLCN-KO cells are massively higher than the levels of WT-TFE3 in the same cells (compare lane 6 with lane 2). Once again, it is impossible to make any comparison between these two samples.

Fig 6D: The authors need to compare on the same blot the phosphorylation of TFE3 fusion proteins in WT and FLCN-KO cells with the phosphorylation levels of WT TFE3 in the same cells. Are these fusion proteins efficiently phosphorylated compared to WT TFE3 in control cells? Is there any difference in the phosphorylation of WT vs chimeric TFE3 (expressed at similar levels) in FLCN-KO cells? Without this comparison it is impossible to assess if there is any "rescue" of TFE3 phosphorylation by linking it to specific cellular compartments in FLCN-KO cells.

Fig6B-C: the localization of TFE3-Rheb15 appears diffuse with no evident colocalization of TFE3 with either lysosomes, Golgi or ER. Similarly, I cannot appreciate any co-localization between TFE3-Tmem192 with LAMP2 (lysosomes), or between TFE3-Sec61b and PDI (ER). I suggest to perform higher quality IF analysis for co-localization of the chimeric protein with the specific cellular compartments and to perform, in addition, biochemical fractionation of the different organelles to confirm these data.

9) Related to point 7), the title of the paragraph "RagC, but not RagA, rescues TFE3 phosphorylation in the absence of FLCN" (lane 168) is misleading. The authors may want to change it with "Active RagC rescues TFE3 phosphorylation in the absence of FLCN, while active RagA does not".

10) The levels of total TFE3 in Fig 1A seem to vary considerably in the different treatments/cell lines. The authors should quantify their data and the levels of pTFE3 should be normalized with total TFE3.

11) Figure 3: In panel A the authors should clearly label WT and FLCN-KO cells; in panel B a staining for RagA/C should be used to show which cells express active RagC/A. Panel C: Due to the huge difference in the expression levels of TFE3 targets in WT and FLCN-KO cells, the graph in Fig 3C is hard to visualize. The transcript levels of TFE3 targets should be represented as fold change of mRNA levels in FLCN-KO cells VS NTC or the authors should separate the data of the two cell lines to allow a better comparison.

12) Figure legends lack the information regarding the statistics for each experiment and the number of times (n) experiments were performed. In addition, scale bars are missing in Fig 1, Fig 2, Fig 3, Fig 5 and Fig 6.

Rev. 2:

It was previously shown that mTORC1-dependent phosphorylation of the transcription factor TFE3 is lost in FLCN deficient cells. Here, the authors study the mechanism of TFE3 phosphorylation by mTORC1. In response to amino acids, FLCN activates RagC, which in turn recruits TFE3 to the lysosome. The authors show that in FLCN or RagC deficient cells, TFE3 is active in the nucleus. This phenotype is suppressed by overexpression of constitutively active RagC. Phosphorylation of the mTORC1 substrate S6K is unperturbed in FLCN or RagC deficient cells. Finally, the authors claim that by expelling TFE3 from the nucleus, by forcing localization to the lysosome, ER, Golgi or cytoplasm, TFE3 phosphorylation is restored even in the absence of FLCN. Specific comments are as follows.

1) The authors study TFE3 phosphorylation in cells cultured in different media: full medium, amino acid deficient, serum deficient and lacking both amino acids and serum. Based on this, the authors claim that growth factor signaling is dispensable for TFE3 phosphorylation (lines 105 -106 and 134), despite the fact that mTORC1 is normally growth factor dependent. Nevertheless, in Fig.1A TFE3 phosphorylation is reduced in serum starved cells compared to full medium. Also, surprisingly, in TSC2 deficient cells, TFE3 phosphorylation is reduced in all conditions. How do the authors explain these phenotypes? The authors should provide an explanation for the loss of TFE3 phosphorylation upon serum starvation (if mTORC1 is not the TFE3 kinase). They should also provide a more compelling explanation for why TFE3 phosphorylation is reduced in TSC2 deficient cells. In general, there are contradictory and confusing claims on the role of mTORC1 in TFE3 phosphorylation.

2) The experiments were performed exclusively in C2C12 cells. Do their findings apply to other cell lines?

3) Fig. 6. The authors force TFE3 localization to the cytoplasm, ER, Golgi or lysosome. Under these conditions, the authors claim that TFE3 is phosphorylated by mTORC1 even in the absence of the mTORC1 activator FLCN. How do the authors explain this observation?

4) Regarding the concluding statement in lines 341-342, the author claim to "elucidate the mechanistic basis by which FLCN confers substrate specificity upon the mTORC1." This statement is too strong. There are still open questions on the exact regulation (see point 1). Moreover, the authors fail to mention a publication from the Ballabio group (G. Napolitano et al, Nature, 2020) claiming a substrate-specific mechanism of TFEB phosphorylation by mTORC1. The authors should rephrase their concluding statement.

5) The authors claim that S6K phosphorylation is unperturbed by deletion of the FLCN gene. However, it has been shown that in FLCN deleted cells S6K phosphorylation is either lost or increased compared to control (e.g. Z. Tsun et al, Molecular Cell, 2013 and Y. Hasumi et al, Human Molecular Genetics, 2014). How can these different phenotypes be explained by the authors' findings? On a similar note, the authors claim that RagC deletion has no effect on S6K phosphorylation. However, the James group (e.g. G. Yang et al, EMBO, 2018) showed a reduction in S6K phosphorylation in RagC shRNA treated cells. The manuscript would benefit from inclusion of these two points the Discussion.

6) Related to statement of lines 196-197, the authors use Torin to increase intracellular amino acids level while blocking mTORC1 activity. The authors should explain how Torin, an mTORC1 and mTORC2 inhibitor, increases nutrients in cells.

7) Fig. 3C. Graphs in this image lack error bars.

8) Introduction. The Introduction would benefit from more citations. Lines 48-52 and 56-64 completely lack references where some should be provided.

9) Figures. To improve clarity, the authors should always add treatment information (e.g., drug concentration, time and media condition) in figure legends. Figures should be self-explanatory. For example, in Fig. 6 the authors should indicate which gene is KO'ed (FLCN). Moreover, it is difficult to distinguish colors in dot bar graphs. The authors should make graphs clearer. Finally, the authors should add axis labels to Fig. 1C and Fig. 2C.

10) The blots are generally over-exposed which might mask important differences. The total TFE3 antibody gives a different pattern in Fig 2A.

---

## [Decision Letter · Decision Letter 2]

1 Dec 2021

Dear Dr Arany,

Thank you for submitting a revised version of your manuscript "FLCN promotes substrate-selective mTORC1 activity by activating RagC to recruit TFE3" for consideration as a Research Article at PLOS Biology. This revised version of your manuscript has been evaluated by the PLOS Biology editors, the Academic Editor and the two original reviewers.

As you will see, both reviewers agree that most of the concerns raised in the previous round have been addressed, but they still mention several issues that need to be confirmed or clarified. One of the main points is the conclusion that S6K can be phosphorylated by mTORC1 in the absence of RagC/D, which is not convincingly demonstrated. Reviewer 1 thinks that you need to perform experiments to show if RagA/B can activate mTORC1 in the absence of RagC/D, and how RagA/B monomers activate mTORC1. The reviewer also suggests several other experiments to confirm your findings and the clarification of some points. Reviewer 2 raises similar issues and asks for several missing controls. After consulting with the Academic Editor and the rest of the team, we have decided to invite you to submit a new revision that should address all the remaining points highlighted by the reviewers. In addition, please remember to add references and make accurate attributions to previous work in the field.

In light of the reviews (attached below), we are pleased to offer you the opportunity to address the remaining points from the reviewers in a revised version that we anticipate should not take you very long. We will then assess your revised manuscript and your response to the reviewers' comments and we may consult the reviewers again.

We expect to receive your revised manuscript within 1 month.

**IMPORTANT - SUBMITTING YOUR REVISION**

3. Resubmission Checklist

a) *PLOS Data Policy*

b) *Published Peer Review*

Sincerely,

Ines

--

Ines Alvarez-Garcia, PhD

Senior Editor

PLOS Biology

Reviewers' comments

Rev. 1:

The authors addressed most of my previous concerns. However, the following important issues remain unsolved:

1. INDEPENDENCE OF S6K PHOSPHORYLATION FROM THE PRESENCE OF RAGC/D

Based on the data shown in new Supplementary Figure 3, the authors claim that S6K can be phopshorylated by mTORC1 in the absence of RagC/D. However, as I had already pointed out in the first revision of this manuscript (point 7), a clear distinction should be made between the presence and the activity of Rag C/D GTPases. While the authors convincingly show that the activity of RagC/D is dispensable for the phosphorylation of S6K, the data shown in this manuscript are too weak to claim that S6K can be phosphorylated in the absence of RagC/D. Furthermore, there is a vast and robust literature showing that Rag GTPases can only work as heterodimers of RagA/B bound to RagC/D. No evidence so far has ever shown or suggested that RagA/B can work as monomers. The possible discovery that RagA/B can activate mTORC1 in the absence of RagC/D would be a revolutionary finding in the field and, therefore, would need to be unequivocally demonstrated with biochemical, structural, and functional analyses. If true, this would be the focus of an entirely different manuscript in which the authors would need to show how RagA/B monomers are able to activate mTORC1. In any case, I remain very skeptical about this possibility as I find the data shown in this manuscript too weak to make such a claim. Therefore, I suggest removing the data on RagC depletion and to keep the data obtained using active RagC, which explore the activity, rather than the presence, of RagC.

2. INDEPENDENCE OF TFE3 PHOSPHORYLATION FROM THE ACTIVITY OF RAGA/B

In response to my concerns in point 6 of the rebuttal, the authors state that they "do not conclude that activation of RagA is dispensable for the phosphorylation of TFE3" and that "RagA is not sufficient to confer activity on TFE3, but it may well be (and likely is) necessary". While I agree with these statements, the authors also state several times in the manuscript that the activity of RagC is sufficient to promote TFE3 phosphorylation. These two concepts contradict each other. The concept that RagC activation is sufficient to induce TFE3 phosphorylation of course implies that the activity of RagA is dispensable, which is in contrast with the authors' own data and statements. Therefore, the authors should fix these issues and re-phrase their claims about the role of the activity Rag GTPases, specifying that while the activity of RagA is essential for the phosphorylation of both TFE3 and S6K, the activity of RagC is important only for the phosphorylation of TFE3.

3. CYTOPLASMIC LOCALIZATION OF TFE3 IS SUFFICIENT TO RESCUE ITS PHOSPHORYLATION IN THE ABSENCE OF FLCN

The authors failed to address my previous concerns on Figure 6. The data contained in this Figure are too weak to support the authors' claim. Figure 6 contains several issues (e.g. a huge difference in the expression of the different proteins analysed, lack of proper localization of the chimeric proteins, inconsistency of data among different panels) and no conclusions can be drawn from these data. Importantly, no mechanism is provided to explain why TFE3 phosphorylation would be rescued in FLCN-KO by simply forcing its localization to different cytoplasmic compartments. This message is in sharp contrast with the overall message of the manuscript (i.e. inactivation of RagC drives TFE3 de-phosphorylation in FLCN-KO cells). Finally, forcing the localization of TFE3 in several cytoplasmic structures (such as Golgi, ER etc…) does not address the "recruitment of TFE3 to mTORC1", as stated in the title of this section. Thus, in my opinion this section, including Figure 6, should be removed from the manuscript.

Rev. 2:

The authors addressed all concerns raised previously. Specific comments are as follows:

1) Related to previous comment 1. Lines 136-139 (before 132-135) state "Together, these data demonstrate that the presence of AAs is necessary and sufficient to promote phosphorylation of TFE3 by mTORC1, and does so via GATOR1 and FLCN, while growth factors are largely dispensable, in sharp contrast to canonical phosphorylation of S6K." The authors should rephrase as this sentence is misleading. The authors should instead include part of the explanation provided to the reviewer, i.e., that the absence of TSC2 is not sufficient to promote TFE3 phosphorylation.

According to the Supplementary Figure 1A, in untreated cells TFE3 phosphorylation is reduced by over 60% in serum-starved conditions compared to full media conditions, indicating at least a partial regulation of TFE3 phosphorylation by growth factors. If the authors want to conclude that S6K phosphorylation dependents more on growth factors than TFE3 phosphorylation, a quantification of S6K phosphorylation of Figure 1A (similarly to Supplementary Figure 1A) should be provided.

2) Regarding comment 2. The authors should explain in the text why in amino acids-starved 293T cells the expression of RagA-66L is not sufficient to promote S6K phosphorylation as shown in Figure 2A for c2c12 cells. Moreover, the author should repeat the experiment in Supplementary figure 2A with equal expression of RagC-WT and RagC-75L. Finally, Figure 2A and Supplementary Figure 2A have different annotations (GDP/GTP vs. 66L/75L), the authors should correct it and keep the same nomenclature in all figures.

3) Supplementary Figure 3C and lines 232-234. The authors' conclusion is not supported by the data. The authors should provide quantification as it appears that phosphorylated S6K is increased upon RagC and RagC/D RNAi treatment. Also, the author should blot phospho and total TFE3 for the same condition (and quantify) as well as RagC and RagD to confirm proper knock down.

4) Regarding comment 5. It would be helpful if the authors included in the Discussion at least part of the explanation provided to the reviewer.

5) Regarding comment 8. The author should add references for lines 48-52 and 56-63.

6) Regarding comment 9. It would facilitate the readers' understanding if the authors indicate which gene is KO'ed (FLCN) in the following figures: 3A, S3A, S4A (both blots), S4B, S5A, S5B (all three blots), 6A, 6C and 6D

---

## [Decision Letter · Decision Letter 3]

8 Feb 2022

Dear Dr Arany,

Thank you for submitting the new revised version of your Research Article entitled "FLCN promotes substrate-selective mTORC1 activity by activating RagC to recruit TFE3" for publication in PLOS Biology. I have obtained advice from one of the original reviewers and have discussed these comments with the Academic Editor. 

Based on the review (attached below), we will probably accept this manuscript for publication, provided you satisfactorily address the two remaining points raised by this reviewer - mainly the removal of Figure 6 and further clarifications on the text. In addition, we have checked carefully the Introduction and we are still not satisfied with the accuracy of some of the statements mentioned regarding previous publications. We think that Reference 13 should be added in the context of line 66 - “We have recently identified a substrate-specific branch of mTORC1 signaling, providing the first example of specific regulation of different branches of mTORC1 signaling [6, 7], subsequently also reported by the Zoncu group [8]” - as the title of that reference states "A substrate-specific mTORC1 pathway". Please also rephrase the last sentence of the Introduction by deleting 'with our input' to avoid misunderstandings: "While the work that we report here was being finalized, the Ballabio group reported overlapping findings with TFEB, with our input [13]".

Please also make sure to address the data and other policy-related requests stated below.

We expect to receive your revised manuscript within two weeks. 

*Published Peer Review History*

*Early Version*

Sincerely,

Ines

--

Ines Alvarez-Garcia, PhD,

Senior Editor,

PLOS Biology

Fig. 1C; Fig. 2B, C; Fig. 3C; Fig. 4A, B; Fig. 5A-C; Fig. 6A-C; Fig. S1A B; Fig. S3B; Fig. S5A and Fig. S6

We require the original, uncropped and minimally adjusted images supporting all blot and gel results reported in an article's figures or Supporting Information files. We will require these files before a manuscript can be accepted so please prepare and upload them now. Please carefully read our guidelines for how to prepare and upload this data: https://journals.plos.org/plosbiology/s/figures#loc-blot-and-gel-reporting-requirements 

Reviewers' comments

Rev. 1:

The authors have satisfactorily addressed some of my criticisms. However, the remaining points are still problematic and need to be properly addressed:

1) As previously stated by this reviewer, data in Figure 6 are very weak and not suitable for publication. The main problems of this figure are: huge difference in the expression of the different proteins analysed, lack of proper localization of the chimeric proteins and inconsistency of data among the different figure panels. Thus, in my opinion Figure 6 needs to be removed from the manuscript.

2) The authors did not properly addressed point 2 of my report: INDEPENDENCE OF TFE3 PHOSPHORYLATION FROM THE ACTIVITY OF RAGA/B. While the authors state that they agree with this reviewer and that RagC is NOT sufficient for TFE3 phosphorylation, as RagA activity is in fact necessary for this process, they now state that "RagC is sufficient to promote TFE3 phosphorylation without simultaneous additional activation of RagA/B". This would imply that Rag GTPases can recruit and phosphorylate TFE3 even when RagA/B are inactive. This is wrong, as it is well established that activation of RagA/B is required for mTORC1 lysosomal recruitment and activation. The text should be changed accordingly.

---

## [Editor Report · Decision Letter 4]

7 Mar 2022

Dear Dr Arany,

On behalf of my colleagues and the Academic Editor, Anne Simonsen, I am pleased to say that we can in principle accept your Research Article entitled "Folliculin promotes substrate-selective mTORC1 activity by activating RagC to recruit TFE3" for publication in PLOS Biology, provided you address any remaining formatting and reporting issues. These will be detailed in an email that will follow this letter and that you will usually receive within 2-3 business days, during which time no action is required from you. Please note that we will not be able to formally accept your manuscript and schedule it for publication until you have any requested changes.

PRESS

Sincerely, 

Ines

--

Ines Alvarez-Garcia, PhD 

Senior Editor 

PLOS Biology
